# QUEST: QUERY-CENTRIC DATA SYNTHESIS APPROACH FOR LONG-CONTEXT SCALING OF LARGE LANGUAGE MODEL

**Chaochen Gao**[1,2], **Xing Wu**[1,2,3 ✉], **Qi Fu**[3], **Songlin Hu**[1,2 ✉]

[1]Institute of Information Engineering, Chinese Academy of Sciences
[2]School of Cyber Security, University of Chinese Academy of Sciences
[3]Xiaohongshu Inc
{gaochaochen,wuxing,husonglin}@iie.ac.cn
fuqi@xiaohongshu.com

## ABSTRACT

Recent advancements in large language models (LLMs) have highlighted the importance of extending context lengths for handling complex tasks. While traditional methods for training on long contexts often use filtered long documents, these approaches lead to domain imbalances, limiting model performance. To address this, techniques like random document concatenation (*Standard*) and similarity-based methods (KNN, ICLM) have been developed. However, they either sacrifice semantic coherence or diversity. To balance both aspects, we introduce Quest, a query-centric data synthesis method aggregating semantically relevant yet diverse documents. Quest uses a generative model to predict potential queries for each document, grouping documents with similar queries and keywords. Extensive experiments demonstrate Quest's superior performance on long-context tasks, achieving remarkable results with context lengths of up to 1M tokens and confirming its scalability across various model sizes.

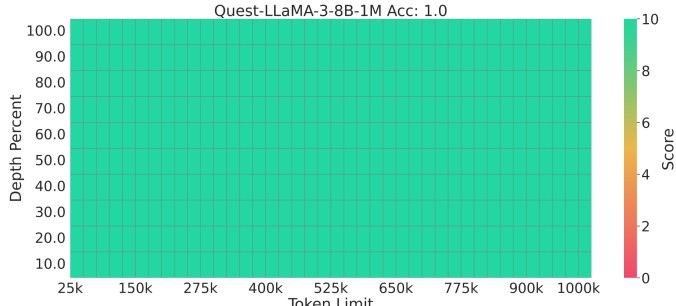

Figure 1: The Needle-in-a-Haystack task evaluates a model's ability to retrieve specific information (the needle) from a large collection of documents (the haystack). Following LongVA (Zhang et al., 2024a) and LWM (Liu et al., 2024), where the x-axis represents the document length and the y-axis indicates the position of the "needle" within the document, ranging from 25K to 1M tokens. To the best of our knowledge, Quest is the first base model (without instruction tuning) to achieve 100% accuracy with a 1M context length.

## 1 INTRODUCTION

Large Language Models (LLMs) are typically pre-trained using fixed context lengths. Recent advancements, however, have highlighted the importance of extending the context lengths. For instance, the LLaMA series has progressively increased its context lengths from 2k tokens in LLaMA to 4k in LLaMA2 and 8k in LLaMA3 (Touvron et al., 2023a;b; Meta, 2024a). LLMs equipped with

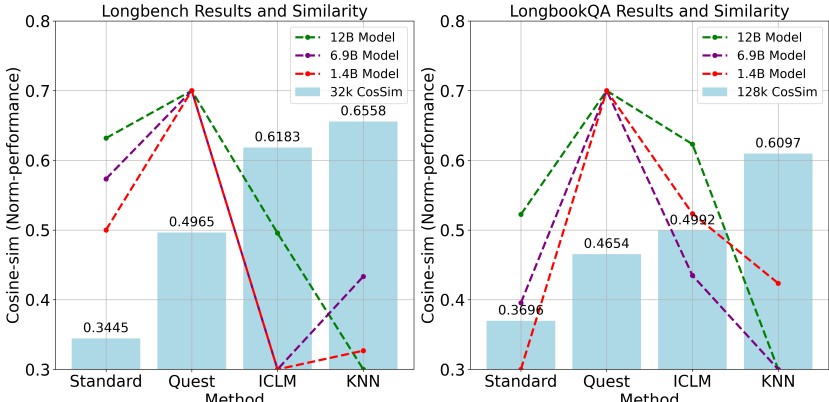

Figure 2: The cosine similarity of aggregated documents and the corresponding performance. The dotted lines indicate the performance of the models, with all results normalized to align within the specified similarity range. High similarity means the semantic correlation is strong, and low similarity indicates good context diversity. Quest balances the semantic correlation and context diversity, resulting in the best performance.

longer context lengths excel at handling complex tasks (Caciularu et al., 2023; Bairi et al., 2023; Mazumder & Liu, 2022), especially those with long input, e.g., document summarization. The goal of long-context modeling is to improve a model's ability to capture long-range dependencies by training on extended contexts. To handle long contexts—such as 128k tokens—a common approach is to continue training LLMs on long-context data (Roziere et al., 2023; Xiong et al., 2023; Fu et al., 2024). Previous works (Xiong et al., 2023; Fu et al., 2024) select such data by filtering long documents from the training set that fit the target context length. However, those documents often come from a few specific domains like Books3 or Arxiv, leading to a skewed distribution, which impacts model performance after continued training (Jung & van der Plas, 2024; Cai et al., 2023).

Previous studies synthesise long-context data by concatenating shorter documents to achieve a balanced domain distribution. Those methods can be classified into two categories: the *Standard* method (Roziere et al., 2023; Ouyang et al., 2022; Le Scao et al., 2023; Touvron et al., 2023a) and similarity-based methods like KNN (Guu et al., 2020; Levine et al., 2021) and ICLM (Shi et al., 2023). The *Standard* method *randomly* concatenates short documents to meet a specified target length. While this approach ensures diversity within the context, the weak semantic relationship between concatenated documents hinders learning long-range dependencies. In contrast, similarity-based methods aggregate semantically similar documents, e.g., by concatenating a document with its top $k$ most similar counterparts from the corpus. However, similarity-based methods overemphasize semantic correlation. They are prone to falling into a narrow context (high redundancy) due to concatenating similar or even repeated documents. Figure 2 compares semantic correlation levels of different methods with their performance on long-context tasks. The results show that either prioritizing context diversity at the expense of semantic correlation (*Standard*) or overemphasizing semantic correlation while sacrificing context diversity (KNN and ICLM) leads to suboptimal performance. Therefore, both context diversity and semantic correlation are crucial for effectively modeling long texts, highlighting the need for a method to balance both aspects.

To achieve the balance effectively, this paper proposes Quest, a query-centric data synthesis approach that simultaneously ensures semantic correlation and context diversity within the long-context data. Our inspiration stems from the observation that similar queries can aggregate semantic relevant but low-redundancy documents via search engines (Mallia et al., 2021; Babenko & Lempitsky, 2014; Kaushik et al., 2004). A straightforward way is to collect enormous queries and cluster articles related to each query, mimicking the functionality of search engines. However, collecting massive queries is very time-consuming, and it is hard to guarantee diversity. To overcome this, we predict potential queries using a generative model for each document in the training dataset. Specifically, Quest begins by employing a lightweight query-prediction model (Raffel et al., 2020; Nogueira et al., 2019; Wu et al., 2022) to predict varied potential queries for each document. Documents sharing the same query are grouped as relevant, simulating an inverse search process.

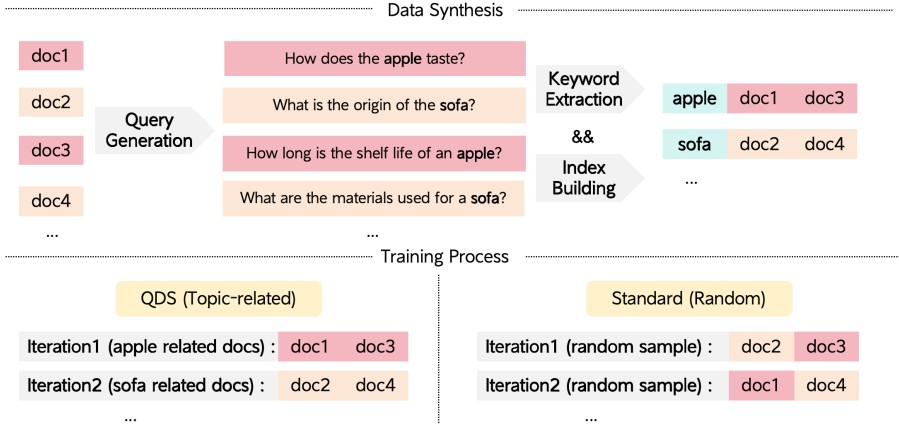

Figure 3: Overview of Query-centric data synthesis (Quest) method. Unlike the *Standard* pre-training strategy that randomly shuffled documents in the input context, Quest places relevant documents in the same context.

To cluster similar queries, we extract more coarse-grained and high-level keywords from queries. Thus, the same keywords further index documents associated with similar queries. Finally, Quest randomly samples from documents indexed by the same keywords and concatenates the selected documents to build long-context data.

Through extensive experiments, we show that Quest significantly outperforms other data synthesis methods on multiple long-context benchmarks with context lengths ranging from 32k to 128k. Figure 1 shows that applying the Quest method to 1M context length achieves impressive performance on the widely used Needle-in-a-Haystack task. Additionally, we further investigate the scaling laws of synthesized long-context data across various model scales and confirm the predictability of the Quest method, making it a reliable solution for advancing long-context models.

Our contributions are summarized as follows:

1. We propose a query-centric data synthesis method to alleviate long-context data scarcity and uneven domain distribution.

2. Extensive experiments on 32k and 128k context lengths show that our method outperforms existing approaches.

3. We provide a detailed analysis of key design elements in Quest and offer valuable insights for future research in the long-context area.

4. We investigate the scaling law of synthesized long-context data and confirm the predictability of our method.

## 2 RELATED WORK

**Long-Context Language Models** The success of LLMs has sparked interest in enabling them to process longer texts. Some works adapt methods for longer texts without additional training by modifying position encoding. For example, Han et al. (2023) and Xiao et al. (2023) adjust the attention matrix to generate long contexts, while Jin et al. (2024) compresses position encoding into the pre-trained position range. Other works involve continued training for better performance. Xiong et al. (2023) demonstrates that long-context capabilities can be acquired by continually pre-training from short-context models. Chen et al. (2023b) uses position interpolation to change the distribution of position encoding, and Yen et al. (2024a) proposes context expansion with parallel encoding. Advances have also been made using RoPE (Su et al., 2021), enabling LLMs to handle longer positions. PoSE (Zhu et al., 2023) employs skip-wise position indices, allowing position encoding to adapt to different lengths. However, those approaches often overlook the scarcity and uneven distribution of long text data during continued training, relying on filtering long documents

from existing corpora (Xiong et al., 2023; Fu et al., 2024; Liu et al., 2024) or randomly splicing short documents to reach a longer length (Roziere et al., 2023; Chen et al., 2023c; Tworkowski et al., 2024; Chen et al., 2023a; Li et al., 2023).

**Data Synthesis and Augmentation for Long-Context**     Acquiring effective long-context data for training is challenging. Some previous retrieval-augmented pre-training works (Guu et al., 2020; Levine et al., 2021) can synthesize long-context data. Guu et al. (2020) clusters semantically similar texts within the same context window, while Levine et al. (2021) shows that incorporating semantically related but non-adjacent sentences within the same pre-training example enhances sentence representations. Shi et al. (2023) uses a traveling salesman algorithm to address document redundancy in the KNN method.

**Scaling Laws**     For a broad spectrum of factors $x$, scaling laws (Kaplan et al., 2020; Henighan et al., 2020; Hoffmann et al., 2022) indicate that their impact on the loss $L$ of a pre-trained model follows a power law relationship. Here, $x$ may represent model sizes, quantities of training data, or training steps, with parameters to be determined. Previous research (Alabdulmohsin et al., 2022; OpenAI, 2023; Bi et al., 2024; Su et al., 2024; Xiong et al., 2024) highlights the impressive predictive power of scaling laws. Notably, fitting this relationship to a set of smaller models, training datasets, or computational resources enables precise extrapolation to predict the test loss for much larger cases across several orders of magnitude. This capability allows practitioners to estimate the performance of a pre-trained larger language model without incurring the substantial cost of completing extensive training runs. However, the scaling law for synthesized long-context data remains unexplored despite its importance for long-context modeling. Therefore, using our Quest method, we investigate the scaling laws of synthesized long-context data across various model sizes and confirm the predictability of the Quest method.

## 3  METHOD

This section details our proposed Query-centric data synthesis (Quest) method. Algorithm 1 presents the method for synthesizing long-context data. Given a dataset with diverse documents $D = \{d_i\}$, our goal is to effectively aggregate relevant but low-redundancy documents for synthesizing training texts with a context length of $L$. An overview of our approach is illustrated in Figure 3. Quest mainly includes 5 steps.

---

**Algorithm 1** Query-centric Data Synthesis (Quest) Method

**Require:** Dataset $D = \{d_i\}$, Context length $L$, Split ratio $r$
**Ensure:** Training texts with context length $L$
 1: Initialize lists $Q$ and $K$
 2: **for** each $d_i \in D$ **do**
 3:     $Q \leftarrow Q \cup \text{doc2query}(d_i)$
 4: **end for**
 5: **for** each $q_i \in Q$ **do**
 6:     $K_i \leftarrow \{k \in \text{Rake}(q_i) \wedge \text{score}(k) \geq 3.0\}$
 7:     $K \leftarrow K \cup \{\text{random}(K_i)\}$
 8: **end for**
 9: $I \leftarrow \{(k_i, d_i) \mid d_i \in D\}$
10: Sort $I$ by size and split: $I_s = \{i \in I \mid \text{rank}(i) \leq r \times |I|\}$, $I_l = I \setminus I_s$
11: **for** each training step **do**
12:     Sample $I_k \in I_s \cup I_l$ (oversample $I_s$)
13:     $T \leftarrow \text{concat}(\text{sample}(I_k)), |T| \geq L$
14:     Train with $T$
15: **end for**

---

1.  **Query Prediction:** We utilize the open-source doc2query model (Nogueira et al., 2019) to predict $n$ queries $\{q_i\}$ for each document $\{d_i\}$. We segment texts that exceed the context length limit of the doc2query model into parts and generate a query for each segment. Consequently, for a document $\{d_i\}$, a list of queries $Q_i = \{q_i^1, \ldots, q_i^n\}$ is predicted. Appendix C.2

demonstrates that higher query quality leads to better model performance, while Appendix C.3 shows that query prediction can effectively improve the quality of keywords.

2. **Keyword Extraction:** We extract keywords from each query $\{q_i\}$ with an efficient tool, *Rake*[1]. For texts with multiple queries, *Rake* generates several lists of keywords $K_i = \{k_i^1, \ldots, k_i^n\}$. To ensure the quality of extracted keywords, we adopt two filtering strategies. First, we filter out keywords with a *Rake* score below 3.0. Second, we remove frequent but non-informative keywords such as "following sentence" or "best way" (see Appendix A.3 for details). Then, we randomly select one of the remaining keywords to serve as the representative keyword for the document. Appendix C.4 presents ablation studies on different methods for selecting keywords, demonstrating that randomly selecting keywords can improve both keyword diversity and model performance.

3. **Building a Keyword-based Inverted Index:** We then build a keyword-based inverted index $I$ after we map each document to its representative keyword. Documents with an identical representative keyword are indexed together and treated as topically similar ones.

4. **Indexes Split:** We found that the number of documents associated with different keywords varies significantly. To address that imbalance and achieve a more balanced data distribution, we implement oversampling for documents with less frequent keywords. Specifically, we rank the keywords in ascending order based on the number of documents indexed by each keyword and divide the sorted keywords into two sets. The top-ranked *split_ratio%* of the keywords are denoted as the short-index set $I_s$, while the remainder is denoted as the long-index set $I_l$. Appendix A.4 shows the impact of *split_ratio%*.

5. **Training Process:** We perform sampling without replacement from the documents within a sampled keyword and concatenate the selected documents up to the target context length $L$ for training. We oversample the short-index set to ensure that the number of tokens participating in training is evenly distributed between the keywords in both $I_s$ and $I_l$. Oversampling redistributes the sampling probability to prioritize $I_s$, ensuring it gets a larger share of the total samples. We provide the mathematical formula for oversampling: Assuming $I_s$ contains $n_s$ training samples, $I_l$ contains $n_l$ training samples, $p$ is the oversampling probability, and the total number of training samples to be used is $N$. The number of samples drawn from $I_s$ can be calculated as:

$$I_s = \left\lceil \left( \frac{n_s}{n_s + n_l} + p \right) \cdot N \right\rceil \tag{1}$$

The number of samples drawn from $I_l$ can be calculated as:

$$I_l = N - I_s \tag{2}$$

## 4 EXPERIMENTS

In this section, we first introduce the experimental settings (Section 4.1). Then we provide a detailed description of our baseline methods (Section 4.2) and the experimental results (Section 4.3 and Section 4.4).

### 4.1 EXPERIMENTAL SETUP

We conduct continued training on Pythia (Biderman et al., 2023) models of different scales, specifically 1.4B, 6.9B, and 12B. Pythia is a series of models trained on the Pile (Gao et al., 2020) dataset, explicitly designed for research. Experiments conducted with Pythia offer good reproducibility. We use identical training sets for all methods to ensure a strictly fair comparison. The only variation between the methods is how to rearrange documents into long-context data.

We apply the Quest method on Pythia's pre-training data, i.e., the Pile dataset, which does not lead to domain transfer issues. Specifically, we extract 30B tokens of keyword-indexed documents from the 300B tokens of the original Pile dataset. Documents indexed by an identical keyword are randomly concatenated to form long-context data that reach the training context length.

---

[1]https://pypi.org/project/rake-nltk

Table 1: Comparison of Longbench results across methods. "Avg." represents the average over multiple test sets. The proposed Quest consistently outperforms baseline methods across various model sizes in the 32K context length setting. Detailed results can be found in Appendix B.

| Train&Test | Model size | Method | Avg. | Sgl. | Multi. | Sum. | Few. | Syn. | Code. |
|---|---|---|---|---|---|---|---|---|---|
| 32k | 1.4B | *Standard* | 20.94 | 19.24 | 17.46 | 20.65 | 26.75 | 2.04 | 36.41 |
|  |  | KNN | 19.97 | 17.26 | 13.01 | **22.97** | 24.16 | 2.33 | 39.22 |
|  |  | ICLM | 19.82 | **20.01** | 14.71 | 21.95 | 23.09 | 1.94 | 35.31 |
|  |  | Quest | **22.06** | 17.97 | **17.98** | 21.91 | **28.06** | **2.33** | **42.25** |
| 32k | 6.9B | *Standard* | 22.48 | 18.07 | **16.83** | 22.33 | **30.23** | **3.86** | 40.91 |
|  |  | KNN | 21.65 | 18.5 | 13.64 | 22.56 | 28.18 | 3.76 | 41.88 |
|  |  | ICLM | 20.86 | 17.82 | 15.34 | 22.35 | 25.86 | 1.21 | 41.15 |
|  |  | Quest | **23.23** | **19.21** | 14.13 | 22.45 | 30.14 | 2.96 | **50.55** |
| 32k | 12B | *Standard* | 24.85 | 22.18 | 21.94 | 22.30 | **32.05** | **3.78** | 43.73 |
|  |  | KNN | 22.95 | 20.55 | 20.48 | 23.51 | 29.19 | 2.47 | 37.44 |
|  |  | ICLM | 24.07 | 22.67 | **23.29** | 23.41 | 30.99 | 1.5 | 37.09 |
|  |  | Quest | **25.24** | **22.34** | 21.08 | **23.74** | 31.91 | 3.22 | **46.8** |

Table 2: Comparison of Longbook QA results across methods. The proposed Quest outperforms baseline methods across various model sizes in the 128 context length setting.

| Train&Test | Model size | Method | Longbook QA |
|---|---|---|---|
| 128k | 1.4B | *Standard* | 9.94 |
|  |  | KNN | 10.36 |
|  |  | ICLM | 10.70 |
|  |  | Quest | **11.30** |
| 128k | 6.9B | *Standard* | 14.47 |
|  |  | KNN | 13.38 |
|  |  | ICLM | 14.92 |
|  |  | Quest | **17.95** |
| 128k | 12B | *Standard* | 17.81 |
|  |  | KNN | 16.42 |
|  |  | ICLM | 18.44 |
|  |  | Quest | **18.92** |

During training, we use the open-source framework GPT-NeoX[2] with a batch size of 4M tokens for all settings. The AdamW optimizer (Loshchilov & Hutter, 2017) with $\beta_1 = 0.9$ and $\beta_2 = 0.95$ parameters and a cosine learning rate schedule is employed. To optimize memory and performance, We use Flash Attention2 (Dao, 2023) and ZeRO (Rajbhandari et al., 2020). The learning rates are $5e^{-5}$ for the 1.4B model, $4e^{-5}$ for the 6.9B model, and $2e^{-5}$ for the 12B model. For more details, refer to Appendix A.

## 4.2 BASELINES METHODS

We compare the proposed Quest method with the existing remarkable data synthesis methods:

1. *Standard* **Method** shuffles and concatenates documents randomly in the input context and has been the mainstream practice in pre-training (Ouyang et al., 2022; Le Scao et al., 2023; Touvron et al., 2023a).

2. **KNN (Retrieval-augmented Language Model Pre-training)** (Guu et al., 2020; Levine et al., 2021) places each document along with the top $k$ most similar retrieved documents in the same input context.

3. **ICLM (Shi et al., 2023) Method** is a recently proposed method that utilizes a traveling salesman algorithm to alleviate the document redundancy problem in the KNN method by ranking similarities and determining the optimal training path.

---

[2]https://github.com/EleutherAI/gpt-neox

Table 3: Comparison of short text performance across methods. Overall, Quest shows almost no degradation in short-text performance on average.

| Model | Avg | Win | PIQA | LogiQA | LAMBADA | Hella | ARC-E | ARC-C |
|---|---|---|---|---|---|---|---|---|
| Pythia | 0.4830 | 0.5746 | **0.7095** | 0.2120 | 0.6163 | **0.4042** | **0.6048** | **0.2594** |
| + *Standard* | 0.4802 | 0.5675 | 0.6975 | 0.2227 | 0.6507 | 0.3943 | 0.5821 | 0.2466 |
| + KNN | 0.4769 | 0.5651 | 0.7089 | 0.2028 | 0.6480 | 0.3946 | 0.5737 | 0.2449 |
| + ICLM | 0.4816 | **0.5785** | 0.7024 | 0.2120 | **0.6546** | 0.3941 | 0.5753 | 0.2543 |
| + Quest | **0.4831** | 0.5691 | 0.7024 | **0.2304** | 0.6472 | 0.3961 | 0.5770 | **0.2594** |

To implement KNN, we utilize a product quantized inverted file (IVFPQ) FAISS index with a code size of 32. For ICLM, we follow the GitHub repository[3] to synthesize long-context data.

## 4.3 EVALUATION RESULTS

We evaluate four methods, including Quest and three baseline methods, with evaluation lengths ranging from 32k to 128k. To comprehensively compare Quest with baseline methods, the datasets from different evaluation tasks are divided into two categories: long-text benchmark and short-text benchmark.

1. **Long-text Benchmark**: For 32k context length, we adopt the widely-used Longbench (Bai et al., 2023) Benchmark, testing six task types: Single-document QA (Sgl.), Multi-document QA (Multi.), Summarization (Sum.), Few-shot learning (Few.), Synthetic (Syn.), and Code completion (Code.), over 17 datasets. For 128k context length, following Fu et al. (2024), we focus on the widely-used Longbook QA task (Zhang et al., 2024b), on which the pre-trained models perform reasonably well without instruction tuning.

2. **Short-text Benchmark**: To assess the effectiveness maintainence of long-text models on short-text tasks, we select 7 widely-used short-text datasets: WinoGrande (Sakaguchi et al., 2021), PIQA (Bisk et al., 2020), Logiqa (Liu et al., 2020), Lambada (OpenAI) (Paperno et al., 2016), HellaSwag (Zellers et al., 2019), ARC-Easy, and ARC-Challenge (Clark et al., 2018).

**Quest demonstrates better performance on average.** Table 1 compares the Longbench results, showing that Quest outperforms other methods on average, especially on the Code dataset. We attribute it to Levenshtein distance, a more stringent and discriminative metric than F1 score or ROUGE, as it accounts for character sequence. To further assess the efficacy of the Quest method in extended long-context settings, we extend the context length to 128k and evaluate the trained models on the Longbook QA task. Table 2 shows that Quest consistently outperforms other methods across various model sizes.

The *Standard* method performs well in 32k tasks but shows the poorest performance on the 128k task. Table 1 shows that the *Standard* method demonstrates good performance in 32k tasks and is superior to KNN and ICLM, especially in few-shot tasks. However, in Table 2, data synthesis methods outperform the *Standard* method, proving the importance of long-context modeling. This improvement can be attributed to the abundance of 32k-length documents, whereas 128k-length documents have a far lower count and uneven domain distributions.

**Quest retains good performance on short text.** To verify how well Quest maintains model performance on short text tasks, we evaluate it on 7 commonly reported tasks, as shown in Table 3. Compared with the base model, the model trained with Quest long-context data shows almost no degradation in short-text performance on average.

## 4.4 APPLYING QUEST ON THE STATE-OF-THE-ART MODEL.

To further verify the effectiveness of Quest, we experiment it with the current state-of-the-art (SOTA) open-source model, i.e., LLaMA3 (Meta, 2024a). Following Fu et al. (2024), we evaluate the Quest-LLaMA3-8B model on the widely used Needle-in-a-Haystack task [4] and Longbook QA task. There

---

[3]https://github.com/swj0419/in-context-pre-training
[4]https://github.com/gkamradt/LLMTest_NeedleInAHaystack

Table 4: Comparisons with the state-of-the-art long-context pre-trained models on the Longbook QA task. Quest-LLaMA-3-8B-128k achieves the best performance among open-source models, surpassing LLaMA-3.1-8B-base and only inferior to the remarkable GPT-4-Turbo-128k. ◇: results from Fu et al. (2024);♣: results evaluated by us.

| Method | Model size | Test Len | Longbook QA |
|---|---|---|---|
| GPT-4-Turbo-128k◇ | - | | **37.40** |
| LLaMA-3-8B (Meta, 2024a)♣ | 8B | | 13.87 |
| LongLoRA (Chen et al., 2023c)◇ | 7B | | 24.30 |
| LongLoRA (Chen et al., 2023c)◇ | 13B | | 24.60 |
| YaRN Mistral (Peng et al., 2023)◇ | 7B | | 26.30 |
| Yi-9B-200K (AI et al., 2024)♣ | 9B | 128k | 30.35 |
| LLaMA-2-7B-80K (Fu et al., 2024)◇ | 7B | | 27.40 |
| DeepSeek-V2-Lite (DeepSeek-AI, 2024)♣ | 16B | | 21.56 |
| Qwen2.5-7B (Team, 2024)♣ | 7B | | 16.57 |
| LLaMA-3.1-8B-base (Dubey et al., 2024)♣ | 8B | | 30.11 |
| Quest-LLaMA-3-8B-128k(ours)♣ | 8B | | **32.39** |

Table 5: Performance comparison of using only the existing long documents in the pretraining corpus versus Quest-synthesized long-context data. "Avg." represents the average over multiple test sets. Quest-synthesized long-context data consistently outperforms the existing long documents on Longbench.

| Method | Avg. | Sgl. | Multi. | Sum. | Few. | Syn. | Code. |
|---|---|---|---|---|---|---|---|
| Long document | 21.11 | 19.77 | 15.15 | 22.11 | 24.81 | 2.46 | 41.84 |
| Quest | **22.06** | 17.97 | 17.98 | 21.91 | 28.06 | 2.33 | 42.25 |

are two approaches to evaluate the Needle-in-a-Haystack task: retrieving a text sentence (Fu et al., 2024) for the 128k context length and retrieving a numeric string (Zhang et al., 2024a; Liu et al., 2024) for the 1M context length. As shown in Figure 4, our Quest-LLaMA3-8B achieves a 97% accuracy on the Needle-in-a-Haystack task (retrieving a text sentence), significantly surpassing the previous highest accuracy of 88% (Fu et al., 2024). In addition, we increased the length to 1M, and Figure 1 shows that Quest achieved 100% accuracy within 1M context length (retrieving a numeric string). Table 4 shows that Quest-LLaMA achieves the highest score among open-source models, further narrowing the gap with GPT-4 Turbo under 128K length setting. These results demonstrate that Quest has strong robustness and scalability when dealing with ultra-long context data.

# 5 ANALYSIS

This section provides an in-depth analysis of Quest. Due to the high computational cost of LLM experiments, our ablation studies are conducted with a 32k context length and 1.4B model size, unless otherwise noted.

## 5.1 QUEST'S ADVANTAGE GRADUALLY EXPANDS WITH TRAINING PROGRESS

This section studies the performance trends of the training progress using data synthesized by the Quest method. As shown in Figure 7, on the Longbench benchmark, the Quest method consistently outperforms other data synthesis methods during the whole training process in terms of superior performance. Additionally, the training progress using the Quest method saturates significantly later. In contrast, other methods generally reach performance saturation within the first 40% of the training progress. Such distinct advantages further demonstrate that Quest's long-context dataset is more diverse and beneficial for long-context training.

## 5.2 QUEST BALANCES DOCUMENT SIMILARITY FOR SUPERIOR PERFORMANCE

To explore how document similarity within the same context influences performance, we randomly sample contexts derived from different methods and calculate the similarity among documents grouped within each context. As shown in Figure 2, model performance initially increases and

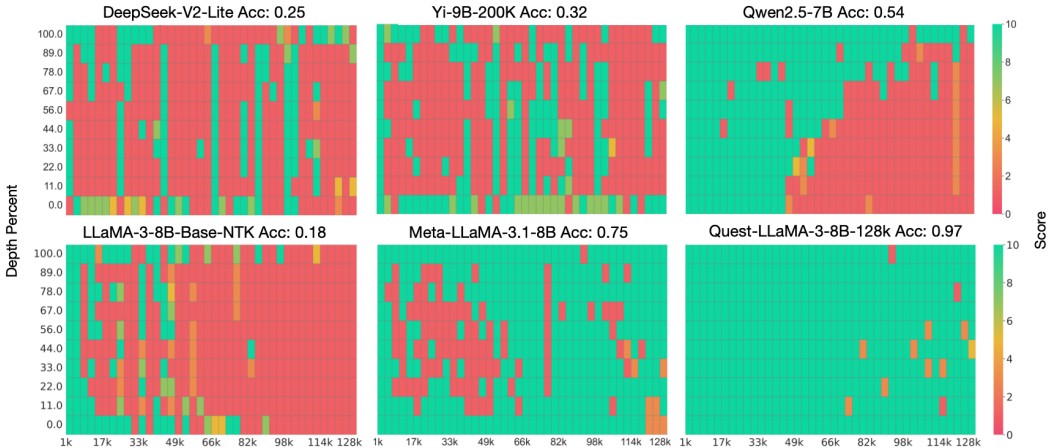

Figure 4: Performance comparison on the Needle-in-a-Haystack task for a collection of base models (without instruction tuning). Quest-LLaMA-3-8B-128k exhibits strong performance, significantly outperforming other open-source models of similar or larger sizes. *Unlike Figure 1, task difficulty is increased within the 128k context length by retrieving a sentence rather than a random numeric string.* "Acc" denotes the percentage of model responses rated as fully accurate (scoring 10 in GPT-4's evaluation) out of all responses generated.

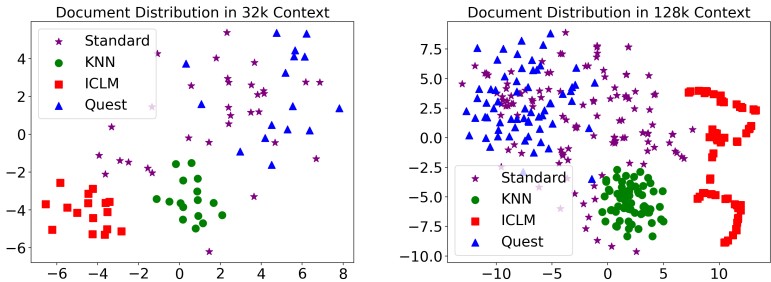

Figure 5: t-SNE visualization of aggregated documents from different methods. The proposed Quest maintains balanced distribution across varying context lengths. See Appendix D.2 for more examples.

then decreases as similarity rises. It indicates that highly similar or irrelevant document aggregations can lead to performance degradation in long-context modeling. When document similarity is excessively high, different documents convey identical information, leading the LLM to repeat previously encountered text for predictions. Conversely, if documents are entirely unrelated, it hinders the model's long-range pattern recognition. Quest achieved the best performance by balancing semantic correlation and context diversity.

We use t-SNE (van der Maaten & Hinton, 2008) to visualize the aggregated documents within the same context. Figure 5 (left) shows that the *Standard* method forms the most dispersed clusters due to random aggregation, while KNN and ICLM yield tighter clusters in the 32k-context-length setting. The proposed Quest exhibits moderate clustering, ensuring semantic consistency and context diversity within the synthesizing long context. Figure 5 (right) illustrates document distribution for various methods under the 128k context length. Quest continues to aggregate relevant, low-redundancy documents.

### 5.3 QUEST-SYNTHESIZED LONG DOCUMENTS OUTPERFORM EXISTING ONES

Some long documents have already reached the target context length in the pre-training corpus. Therefore, we compare the performance of using Quest-synthesized long-context data with using

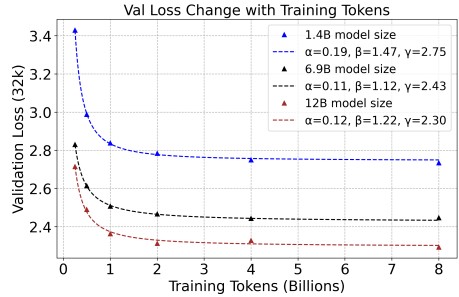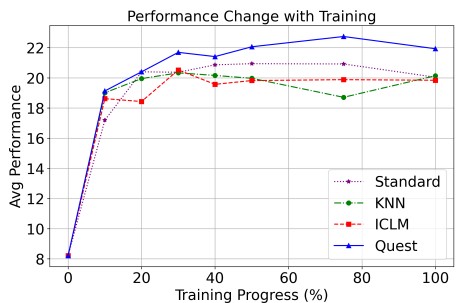

Figure 6: Scaling law of synthesized long-context data under different model sizes. The Quest approach scales well and predictably.

Figure 7: Performance trends during the training progress using data synthesis methods. The Quest consistently outperforms other methods.

only the existing long documents in the pre-training corpus for training. The amount of data in both the synthetic data and the long documents is identical. Table 5 shows that using Quest-synthesized long-context data achieves better results on Longbench. Existing long documents perform worse because they only exist in a few domains, resulting in skewed data distribution, as illustrated in Appendix C.5. The Quest method, on the other hand, can cover nearly all domains, resulting in more diverse synthesized long-context data and better performance in evaluation tasks. We also attempt further comparisons with a context length of 128k. However, long documents exceeding 128k in the Pile dataset are rare and inadequate to support a fair comparison experiment. As the target context length increases, the scarcity problem becomes more pronounced, further highlighting the necessity of developing effective long-context synthesis methods.

## 6 SCALING LAW OF SYNTHESIZED LONG-CONTEXT DATA

To explore the scaling law of synthesized long-context data, we vary the amount of training data for different model sizes (1.4B, 6.9B, and 12B) under the 32k context length setting. Formally, we formulate the scaling law of the validation loss by studying different model sizes $N$ and dataset sizes $D$:

$$L(D) = \alpha \exp(-\beta D) + \gamma$$

This formula applies to each model size, where $\{\alpha, \beta, \gamma\}$ are variables to be learned. In our experiments, each model is trained separately on datasets of different sizes: 250 million, 500 million, 1 billion, 2 billion, and 4 billion tokens. Then, we fit a curve for each model size, showing the relationship between the data scaling and the validation loss at the end of each training, as shown in Figure 6.

We validated the learned scaling law on an 8 billion data size by comparing the relative error between each model's final validation loss and its predicted value. The relative errors were 0.5% for the 1.4B model, -0.5% for the 6.9B model, and 0.4% for the 12B model, demonstrating the scalability and accuracy of Quest's data synthesis approach with minimal deviation.

## 7 CONCLUSION

In this paper, we introduce Quest, a novel method for synthesizing balanced long-context data by grouping and concatenating relevant but low-redundant documents associated with similar queries. The proposed Quest ensures semantic correlation and context diversity in long-context data to improve the long-context modeling capability of pre-trained models. Extensive experiments demonstrate that Quest outperforms existing approaches across various long-context and short-context benchmarks, proving it to be an effective and reliable solution for advancing long-context models.

## 8 ACKNOWLEDGEMENT

This work was supported by the National Natural Science Foundation of China (No. U24A20335).

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

# A  IMPLEMENTATION DETAILS OF QUEST

## A.1  MODEL CONFIGURATION

We present the model configuration in Table 6. For the other baselines, we only altered the training dataset, keeping the default model configuration unchanged.

## A.2  TIME CONSUMPTION

During the dataset construction, query generation utilized 16 H800 GPUs and took 24 hours to process the 30 billion document tokens in The Pile dataset. After that, 128 CPUs were employed for distributed processing: it took 0.5 hours to generate the keywords and 3 hours to construct the inverted index based on those keywords. Since the process is highly parallelizable, scaling it up would not pose significant challenges.

| | 32K | | | 128K | | |
|---|---|---|---|---|---|---|
| model size | 1.4B | 6.9B | 12B | 1.4B | 6.9B | 12B |
| rotary-pct | | | 0.25 | | | |
| rotary-emb-base | | 100000 | | | 5000000 | |
| $\beta_1$ | | | 0.9 | | | |
| $\beta_2$ | | | 0.95 | | | |
| eps | | | $1e^{-8}$ | | | |
| lr | $5e^{-5}$ | $4e^{-5}$ | $2e^{-5}$ | $5e^{-5}$ | $4e^{-5}$ | $2e^{-5}$ |
| precision | | | bfloat16 | | | |
| Zero_stage | | | 1 | | | |
| gradient-clipping | | | 1.0 | | | |
| weight-decay | | | 0.1 | | | |
| lr-decay-style | | | cosine | | | |
| train-iters | | | 1000 | | | |
| warmup-iters | | | 200 | | | |
| seq-length | | 32768 | | | 131072 | |
| GPU-type | | | H800 | | | |
| GPU-numbers | 16 | 32 | 32 | 32 | 32 | 32 |
| training-time | 6.3h | 14h | 20.6h | 9.5h | 30h | 39h |

Table 6: Model Training Configuration

| Column 1 | Column 2 | Column 3 |
|---|---|---|
| best way | get rid | bad idea |
| good way | main differences | valid way |
| following sentence | two sentences | better way |
| mean | passage mean | following data |
| good idea | best ways | correct way |
| sentence mean | next word | following passage |
| part 1 | current state | following equation |

Table 7: Stop Keywords

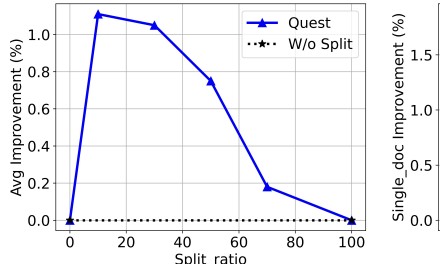

Figure 8: The performance changing trend with *split_ratio* increasing. For detailed results, please refer to Appendix B.

### A.3 FILTERING OF KEYWORDS

We used *Rake* for keyword extraction and found many high-frequency but meaningless keywords. Therefore, we maintained a list of stop words and performed a keyword extraction on the generated query to avoid selecting those stop words. Some of the stop words are listed in Table 7. Moreover, to enhance the quality of keyword extraction, we applied a post-processing step to clean the keywords generated by the *Rake* algorithm. That involves removing punctuation and filtering out keywords that are either less than four characters in length or have a score below 3. Such a cleaning process tends to ensure that the extracted keywords are both meaningful and relevant.

### A.4 IMPACT OF SPLIT RATIO

This section examines the impact of *split_ratio*, which controls the proportion of oversampled keyword-based inverted indexes. Figure 8 shows that performance initially improves but then declines as *split_ratio* increases. The best results occur when oversampled indexes comprise 10-30%. It suggests that moderate oversampling of indexes with fewer documents benefits long-context modeling, highlighting the importance of balanced data distribution for long-context tasks.

Table 8: Performance of different methods across various Longbench subtasks.

| Model Size | Method | Few-shot Learning | | | | Synthetic Tasks | | Code Completion | |
|---|---|---|---|---|---|---|---|---|---|
| | | trec | triviaqa | samsum | nq | passage_count | passage_retrieval_en | lcc | repobench-p |
| 1.4B | *Standard* | 32.29 | 19.99 | 24.75 | 29.95 | 2.47 | 1.62 | 33.66 | 39.16 |
| | KNN | 33.50 | 17.13 | 21.88 | 24.13 | 1.09 | 3.58 | 37.69 | 40.75 |
| | ICLM | 31.75 | 17.91 | 16.08 | 26.60 | 1.00 | 2.88 | 34.56 | 36.07 |
| | Quest | 40.75 | 18.68 | 19.17 | 33.65 | 0.96 | 3.71 | 40.89 | 43.60 |
| 6.9B | *Standard* | 38.75 | 22.38 | 23.06 | 36.72 | 2.97 | 4.75 | 41.17 | 40.66 |
| | KNN | 35.83 | 24.29 | 16.54 | 36.07 | 2.69 | 4.83 | 42.86 | 40.89 |
| | ICLM | 38.08 | 18.90 | 15.47 | 30.97 | 2.15 | 0.27 | 43.37 | 38.93 |
| | Quest | 38.50 | 21.59 | 24.22 | 36.26 | 2.41 | 3.50 | 49.95 | 51.16 |
| 12B | *Standard* | 39.25 | 26.87 | 26.97 | 35.12 | 3.12 | 4.44 | 42.03 | 45.43 |
| | KNN | 39.42 | 21.64 | 20.22 | 35.46 | 2.11 | 2.83 | 36.62 | 38.25 |
| | ICLM | 41.08 | 23.05 | 22.24 | 37.60 | 1.75 | 1.25 | 34.65 | 39.52 |
| | Quest | 38.21 | 26.60 | 21.81 | 41.03 | 2.86 | 3.58 | 46.91 | 46.69 |

Table 9: Performance of different methods across various Longbench subtasks.

| Model Size | Method | Single-Doc QA | | | Multi-Doc QA | | | Summarization | | |
|---|---|---|---|---|---|---|---|---|---|---|
| | | narrativeqa | qasper | multifieldqa_en | hotpotqa | 2wikimqa | musique | gov_report | qmsum | multi_news |
| 1.4B | *Standard* | 13.55 | 14.8 | 29.38 | 21.71 | 22.96 | 7.7 | 23.28 | 14.53 | 24.13 |
| | KNN | 13.29 | 12.24 | 26.24 | 16.72 | 16.33 | 5.98 | 26.29 | 15.31 | 27.32 |
| | ICLM | 14.66 | 15.79 | 29.59 | 16.56 | 20.13 | 7.44 | 25.28 | 14.33 | 26.25 |
| | Quest | 12 | 12.77 | 29.14 | 21.63 | 24.7 | 7.62 | 25.53 | 14.35 | 25.86 |
| 6.9B | *Standard* | 13.83 | 10.78 | 29.61 | 21.66 | 21.52 | 7.31 | 24.08 | 16.66 | 26.25 |
| | KNN | 16.10 | 10.41 | 29.00 | 19.66 | 17.30 | 3.97 | 26.70 | 17.06 | 23.91 |
| | ICLM | 14.62 | 10.65 | 28.19 | 19.48 | 20.44 | 6.11 | 26.08 | 16.41 | 24.56 |
| | Quest | 17.77 | 8.63 | 31.23 | 19.46 | 17.60 | 5.33 | 26.69 | 16.11 | 24.56 |
| 12B | *Standard* | 20.32 | 13.85 | 32.36 | 28.79 | 22.09 | 14.94 | 24.57 | 18.56 | 23.78 |
| | KNN | 17.91 | 12.23 | 31.50 | 24.79 | 23.63 | 13.01 | 28.43 | 18.05 | 24.04 |
| | ICLM | 20.33 | 16.84 | 30.84 | 31.92 | 23.83 | 14.11 | 28.01 | 18.97 | 23.24 |
| | Quest | 19.12 | 14.17 | 33.72 | 27.53 | 21.76 | 13.94 | 28.22 | 19.33 | 23.68 |

Table 10: Performance Change with different $split\_ratio$ values across various Longbench subtasks.

| Split Ratio (%) | Few-shot Learning | | | | Synthetic Tasks | | Code Completion | |
|---|---|---|---|---|---|---|---|---|
| | trec | triviaqa | samsum | nq | passage_count | passage_retrieval_en | lcc | repobench-p |
| 0 | 33.46 | 21.94 | 22.50 | 29.30 | 1.48 | 2.50 | 32.29 | 37.06 |
| 10 | 39.21 | 23.39 | 21.47 | 30.44 | 1.07 | 3.77 | 33.96 | 39.11 |
| 30 | 36.75 | 18.06 | 27.06 | 27.85 | 0.89 | 3.50 | 33.74 | 38.32 |
| 50 | 36.17 | 22.02 | 24.29 | 31.60 | 1.64 | 4.69 | 32.28 | 37.00 |
| 70 | 33.83 | 19.17 | 25.58 | 27.13 | 1.81 | 3.67 | 29.78 | 35.66 |
| 90 | 32.33 | 21.23 | 26.34 | 23.98 | 2.41 | 2.60 | 30.81 | 34.52 |
| 100 | 33.46 | 21.94 | 22.50 | 29.30 | 1.48 | 2.50 | 32.29 | 37.06 |

Table 11: Performance Change with different $split\_ratio$ values across various Longbench subtasks.

| Split Ratio (%) | Single-Doc QA | | | Multi-Doc QA | | | Summarization | | |
|---|---|---|---|---|---|---|---|---|---|
| | narrativeqa | qasper | multifieldqa_en | hotpotqa | 2wikimqa | musique | gov_report | qmsum | multi_news |
| 0 | 12.41 | 13.13 | 27.87 | 20.05 | 16.65 | 8.05 | 24.57 | 15.50 | 22.64 |
| 10 | 13.79 | 16.22 | 27.15 | 18.38 | 18.74 | 9.11 | 26.72 | 15.01 | 22.69 |
| 30 | 14.04 | 17.19 | 27.77 | 20.39 | 20.70 | 7.87 | 26.20 | 14.94 | 23.99 |
| 50 | 13.31 | 13.92 | 27.80 | 18.69 | 21.84 | 7.14 | 24.70 | 15.12 | 23.18 |
| 70 | 11.32 | 14.34 | 28.37 | 17.85 | 22.21 | 9.20 | 25.18 | 14.89 | 24.35 |
| 90 | 13.18 | 18.30 | 29.68 | 19.71 | 21.14 | 7.14 | 24.06 | 15.09 | 23.51 |
| 100 | 12.41 | 13.13 | 27.87 | 20.05 | 16.65 | 8.05 | 24.57 | 15.50 | 22.64 |

Table 12: Performance comparison across different methods on HELMET.

| Method | Avg. | Recall | RAG | ICL | Re-rank | LongQA |
|---|---|---|---|---|---|---|
| *Standard* | 48.08 | 62.33 | **58.67** | 71.24 | 19.18 | 28.99 |
| KNN | 46.31 | 64.24 | 56.00 | 60.28 | 18.77 | 32.27 |
| ICLM | 46.62 | 64.04 | 54.48 | **72.36** | 14.04 | 28.17 |
| Quest | **50.97** | **69.13** | 57.47 | **72.08** | **22.35** | **33.82** |

Table 13: Performance comparison across different query replacement ratios.

| Replacement Ratio | Avg. | Recall | RAG | ICL | Re-rank | LongQA |
|---|---|---|---|---|---|---|
| 0% | **50.97** | **69.13** | 57.47 | 72.08 | **22.35** | **33.82** |
| 20% | 50.33 | 67.35 | 58.07 | 71.72 | **22.55** | 31.98 |
| 50% | 49.74 | 63.96 | 58.19 | **73.32** | 20.93 | 32.31 |
| 100% (*Standard*) | 48.08 | 62.33 | **58.67** | 71.24 | 19.18 | 28.99 |

# B 32K LONGBENCH RESULTS

We report the performance of Longbench on 17 English subtasks. Table 8 and Table 9 are the detailed results of Table 1. Table 10 and Table 11 are the detailed results of Figure 8.

# C ABLATION STUDY IN QUEST

In this section, we conduct evaluations on the long-context benchmark HELMET (Appendix C.1), performing ablations on query quality (Appendix C.2), keyword quality (Appendix C.3), and keyword sampling strategies (Appendix C.4).

## C.1 RESULTS ON HELMET BENCHMARK

We evaluate the Quest method on HELMET benchmark (Yen et al., 2024b). HELMET provides more reliable and consistent rankings for frontier long-context models. It spans evaluation lengths of 8k, 16k, 32k, 64k, and 128k across five task types and 17 subtasks, including Recall (4 subtasks), RAG (4 subtasks), ICL (5 subtasks), Re-rank (1 subtask), and LongQA (3 subtasks). Table 12 represents the average performance across lengths from 8k to 128k. The comprehensive evaluation on HELMET shows that Quest achieves non-trivial improvements across multiple datasets, with a notable +2.89% average increase in performance on as many as 17 subtasks.

## C.2  Query Quality and Model Performance Show a Positive Correlation

We generate queries using large language models (LLMs), which intuitively could produce higher-quality queries. However, the computational cost is prohibitively high and beyond our resource capacity. To explore query quality generation within a manageable range, we simulate the creation of low-quality queries by randomly replacing keywords in the original queries with a certain probability. Table 13 shows that model performance gradually declines as the replacement ratio increases. This observation suggests that improving the quality of queries can further enhance model performance.

## C.3  The keywords from queries are high-quality

We evaluate various strategies for keyword extraction using GPT-4o. Three distinct strategies are designed for generating keywords:

1. **Keywords from Queries:** Extracting keywords from the generated queries.

2. **Keywords from Documents:** Extracting keywords directly from the original documents.

3. **Keywords from Summaries:** Generating summaries from the original documents using a model, then extracting keywords from the summaries. We utilize the recently introduced Llama-3.2-1B (Meta, 2024b) model to generate summaries.

To assess the quality of the extracted keywords, we conduct evaluations using human annotations and GPT-4o on 100 samples. Scores ranged from 1 to 5. Two PhD students are hired for annotation, alongside GPT-4o, all following the scoring criteria below:

Table 14: Performance comparison across different keyword extraction strategies.

| Strategy | Avg Score | Score $\geq$ 3 | Score $\geq$ 4 | Score $\geq$ 5 |
|---|---|---|---|---|
| **Keywords from Queries** | **3.574** | **74.3%** | **63.0%** | **36.4%** |
| **Keywords from Summaries** | 3.009 | 61.8% | 48.7% | 14.5% |
| **Keywords from Documents** | 2.634 | 49.3% | 31.2% | 6.0% |

Table 15: Comparison of entropy scores between keyword selection strategies.

| Strategy | Entropy Score |
|---|---|
| Highest RAKE Score | 15.31 |
| Random Sampling | **18.12** |

Table 16: Performance comparison of keyword selection strategies.

| Method | Avg. | Recall | RAG | ICL | Re-rank | LongQA |
|---|---|---|---|---|---|---|
| Highest RAKE Score | 49.51 | 66.76 | 56.27 | **72.72** | 22.32 | 29.47 |
| Random Sampling | **50.97** | **69.13** | **57.47** | 72.08 | **22.35** | **33.82** |

- **Score 1:** The keyword has little to no relevance to the article, may have been extracted incorrectly, or is significantly off-topic.

- **Score 2:** The keyword has some relevance to the article but is of low importance or overly generalized. It does not represent a core theme or focus of the article.

- **Score 3:** The keyword is clearly connected to the article but is not one of its primary concepts or focal points.

- **Score 4:** The keyword is strongly related to the article and represents an important concept or secondary theme.

- **Score 5:** The keyword perfectly reflects the article's central theme, main concepts, or key discussion points.

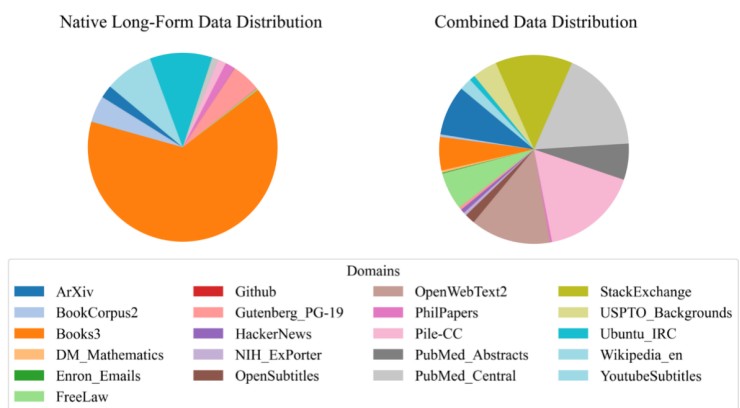

Figure 9: (Left) Distribution of long documents exceeding 128k tokens in the Pile dataset. (Right) Distribution of >= 128k tokens long-context data synthesized by Quest. By employing Quest, we successfully increase the domain diversity and balance of the long-context training set.

Table 17: Performance comparison across KNN strategies and Quest.

| Methods | Cosine-Sim | Avg. | Recall | RAG | ICL | Re-rank | LongQA |
|---|---|---|---|---|---|---|---|
| KNN (TopK) | 60.97 | 46.31 | 64.24 | 56.00 | 60.28 | 18.77 | 32.27 |
| KNN (Mid-Ranking) | 48.53 | 47.96 | 65.71 | 56.97 | 68.24 | 16.54 | 32.33 |
| KNN (Random Sampling) | 35.51 | 46.78 | 64.23 | 55.24 | 70.96 | 12.23 | 31.22 |
| KNN (Reverse Order) | 35.49 | 46.18 | 62.19 | 54.28 | 66.28 | 17.20 | 30.93 |
| Quest | 46.54 | **50.97** | **69.13** | **57.47** | **72.08** | **22.35** | **33.82** |

The Pearson correlation coefficient between human annotations and GPT-4o scores is calculated as 0.7986, indicating a strong alignment between the two evaluation methods.

Subsequently, GPT-4o is used to score 1,000 samples. Table 14 shows that extracting keywords from queries significantly improves the quality of the extracted keywords, and these keywords achieve the highest score. The keyword from Generated summaries is of inferior quality. We attribute this to the broader semantic space of summaries compared to queries, which interfere with effective keyword extraction.

## C.4 RANDOM SAMPLING ENHANCES KEYWORDS DIVERSITY AND IMPROVES MODEL PERFORMANCE

Through entropy analysis, we evaluate the diversity of two strategies: selecting keywords with the highest RAKE score and random sampling (for scores greater than 3). Table 15 shows that random sampling led to greater diversity. Furthermore, we construct a training dataset based on the highest RAKE score, and the results are presented in Table 16. These results demonstrate that random sampling enhances diversity and improves model performance.

## C.5 THE DISTRIBUTION COMPARISON OF LONG CONTEXT DOCUMENTS

Figure 9 shows the distribution of long-context data before and after the application of Quest. While the distribution of long-context sources was initially highly uneven, the implementation of Quest has significantly increased the percentage of data in domains such as ArXiv, FreeLaw, OpenWebText2, Pile-CC, and PhilPapers, where there was previously minimal or no native long-context data.

# D ANALYSIS WITH DOCUMENT SIMILARITY

## D.1 CHANGING THE SIMILARITY OF KNN-AGGREGATED DOCUMENTS CANNOT ACHIEVE QUEST'S PERFORMANCE

As we mentioned in Section 5.2, Quest achieves significant performance improvements by aggregating relevant, low-redundancy documents. This raises the question of whether selecting moderately similar documents instead of the most similar ones in a KNN-based method can achieve comparable results. To perform such similarity in a KNN-based method, we designed four strategies:

1. **TopK:** The normal KNN selects and concatenates documents with top-ranking similarity.
2. **Mid-Ranking:** Documents with mid-ranking similarity are selected and concatenated.
3. **Random Sampling:** Documents are concatenated by random sampling.
4. **Reverse Order:** Documents are concatenated in reverse order of similarity.

As shown in Table 17, for the KNN method, reducing the similarity of the selected documents results in a slight performance improvement, followed by a decline. Furthermore, these methods consistently perform worse than the Quest method. Additionally, while the "Mid-Ranking" approach exhibits a similarity extent close to the Quest method, its performance remains far inferior. It indicates that document aggregation has a more significant impact on overall performance than the extent of similarity, highlighting it as the critical factor driving Quest's superior results.

## D.2 EXAMPLES OF SIMILARITY VISUALIZATIONS.

We present more visualization results. Figure 10 shows the t-SNE visualization of documents within a 32k context, while Figure 11 illustrates documents within a 128k context. The *Standard* method's random concatenation of documents results in an overly dispersed distribution, disrupting document relationships and leading to poorer performance. In a 32k context, ICLM causes excessive clustering due to shorter similarity-path lengths, mirroring a KNN-like distribution and impairing performance on the Longbench benchmark. However, the 128k context allows ICLM to form longer similarity paths, dispersing document distribution and enhancing performance on the LongbookQA benchmark.

Notably, Quest maintains an evenly dispersed document distribution in both contexts, leading to its superior performance on both benchmarks. The findings above indicate that overly dispersed or concentrated document semantics can harm model performance, while Quest improves performance by clustering query-related documents, ensuring relevance and avoiding redundancy.

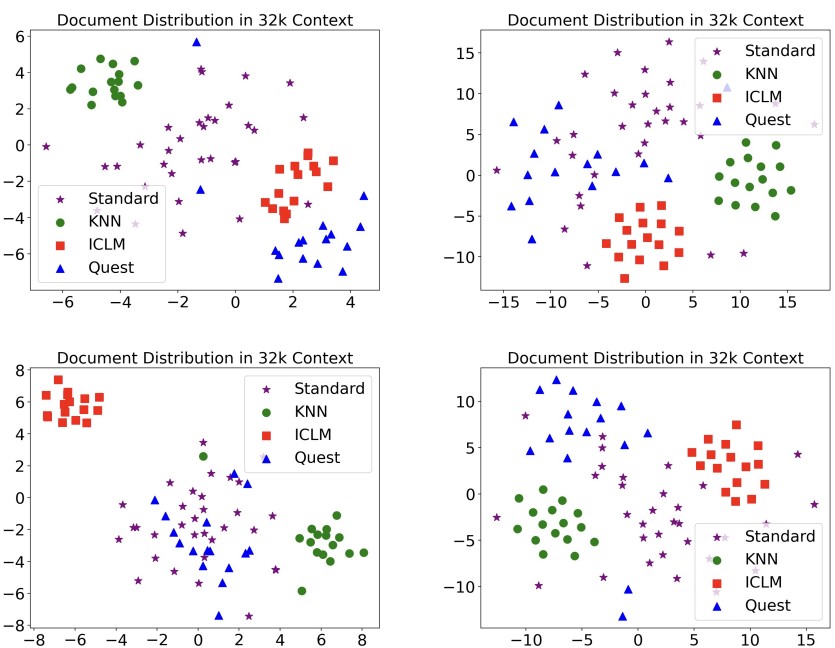

Figure 10: Visualizing Documents Comprising a 32k Context.

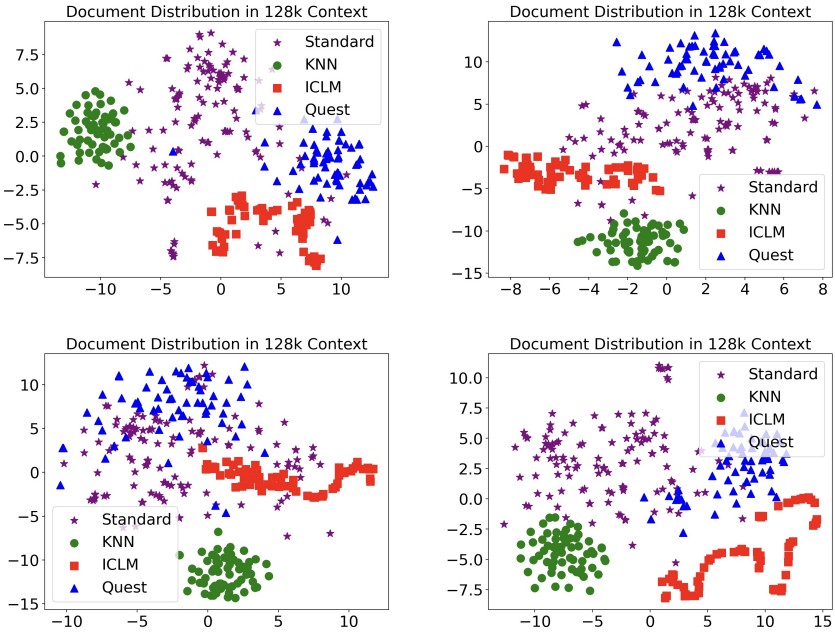

Figure 11: Visualizing s Comprising a 128k Context.

