# OpenReview forum: "Quest: Query-centric Data Synthesis Approach for Long-context Scaling of Large Language Model"
_ICLR.cc/2025/Conference — ICLR 2025 Poster_

### Official Review · Reviewer_YyBr · 2024-10-24

**Soundness:** 3
**Presentation:** 3
**Contribution:** 2
**Rating:** 8
**Confidence:** 4

**Summary:**

This paper introduces Quest, an approach for grouping documents to train long-context LLMs. Existing approaches for document grouping either use random grouping or grouping by similarity. However, the former leads to aggregated documents that contain unrelated pieces, while the latter yields aggregated documents with repeated information coming from similar documents. Quest then seeks to balance the coherence of the aggregated documents and the diversity of their content.

The Quest algorithm first associates each document to a query, which is used to extract a representative keyword associated to the document. Documents are then indexed based on their respective keyword, in the same spirit as an inverted index. During training, a keyword is sampled and an aggregated document is formed by randomly sampling documents associated to this keyword up to the desired context size. The keyword sampling procedure also takes into account the rarity of the keyword (i.e., the number of documents associated to a keyword) by oversampling rarer keywords, to have a more balanced domain representation. Quest is validated through extensive experiments on long-context tasks and is also shown to preserve the short-context capabilities of the original pre-trained model.

**Strengths:**

- The paper is overall well-written, clear and easy to follow.
- The experiments are comprehensive and convincing.

**Weaknesses:**

- Quest is defined as a simple heuristic that is not backed by any theoretical guarantees or motivations.
- Several design choices in the approach lack justifications or intuitive explanations, and thus feel arbitrary (see specific points in the "Questions" section).

**Questions:**

- The first step of the Quest algorithm is to associate documents with keywords. This is done by first formulating a query from the document, before extracting keywords from this query. It's not very intuitive why this document-to-query strategy is favored over applying any standard summarization technique to the document. A summary may retain more of the document's information than a single query, and thus might lead to higher-quality keywords. The paragraph lines 93-105 attempts to make some links with search engines, but the motivation could benefit from being further clarified and strengthened.
- At the end of the keyword extraction step, the keyword that is eventually retained for a document is randomly selected among the different keywords associated to the document. This strategy seems a bit unintuitive and risky, as it could lead to the selection of a keyword that is only marginally relevant to the document. Why isn't the Rake score used here to decide which keyword to retain? Or even simply retain the keyword with the largest Rake score?
- Was there any evaluation conducted to validate the quality of the keywords identified for the documents? This could for example be done by asking some human annotators to provide keywords for a few documents and compare these against the keywords obtained through Quest, or instead ask them to rate the relevance of the Quest keywords. This seems like an important sanity check to conduct given the importance of proper keyword-document matching in the proposed approach.
- How is the oversampling of keywords from the short-index set $I_s$ done in practice? How are the probability of sampling a keyword from $I_s$ and that of sampling from $I_l$ defined, exactly? Adding the mathematical formula for this would be helpful for the reader's understanding.
- What is the motivation for splitting $I$ into only two sets $I_s$ and $I_l$ instead of having a more fine-grained approach? For example, one could define an adaptive probability of sampling a keyword based on the number of tokens in its associated documents. Such an approach would enable a more tailored treatment of the different keywords to compensate for their rarity.
- Figure 3 contains several typos: "query genaration" -> "query generation", "random smaple" -> "random sample"
- In the results on Longbench in Table 1, it seems that the superior average performance of Quest over baselines is essentially due to its significantly better results on the Code Completion task. The good performance of Quest on this specific task is attributed to the use of Levenshtein distance as the metric. It is however not entirely clear why using Quest would specifically boost this metric, so more detailed explanations would be appreciated.
- It is surprising that Quest achieves 100% accuracy on the needle-in-a-haystack task for a context of 1M tokens, but gets 97% accuracy for a context of 128k tokens (which should intuitively be easier). Are there any intuitive explanations for this phenomenon?

---

> ### Author Response · Authors · 2024-11-21
> **Response to Reviewer YyBr**
>
> **Q1: The paragraph lines 93-105 attempts to make some links with search engines, but the motivation could benefit from being further clarified and strengthened.**
>
> **A1:** Thank you for your valuable suggestion. We will further clarify and strengthen our motivation in the latest version. Our primary goal is to collect enormous queries and cluster articles related to each query, mimicking the functionality of search engines. However, collecting many queries is very time-consuming and cannot guarantee diversity. Thus, we employ the doc2query model to generate queries for each document in the corpus. This way, we solve the problem of cheaply acquiring many diverse queries.
>
>
> **Q2: Why isn't the Rake score used here to decide which keyword to retain? Or even simply retain the keyword with the largest Rake score?**
>
> **A2:** In fact, the RAKE score is used to filter out meaningful keyword sets based on our statistical analysis, indicating that keywords with scores greater than 3.0 are meaningful ( As we mentioned in line 213). Then, to further ensure the diversity of documents related to each keyword and mitigate any bias introduced by RAKE, we randomly select from the remaining keywords after filtering.
>
> The rationale behind randomly selecting RAKE keywords is to maintain context diversity. For example, consider two similar documents, document A and document B, which may share keywords like "sofa" and "table." We aim to prevent these documents from being consistently associated with the keyword that holds the higher RAKE score. Instead, we randomly assign each document to one of the keywords, assigning document A to "sofa" and document B to "table." This random assignment ensures a greater diversity of documents linked to each keyword.
>
> **Q3: Was there any evaluation conducted to validate the quality of the keywords identified for the documents?**
>
> **A3:** We first scored the extracted keywords using human annotation and GPT-4o on 100 samples, ranging from 1 to 5. The Pearson correlation coefficient between human annotations and GPT-4o scores was calculated as 0.7986, indicating a strong correlation between the two methods.
>
> Subsequently, We employed GPT-4 to score 1000 samples. The results are presented in the following table:
>
> | **Metric**                           | **Quest keywords** |
> |--------------------------------------|---------------------------|
> | **Average Score**                    | 3.574                     |
> | **Keywords Scoring ≥ 3**         | 74.3%                     |
> | **Keywords Scoring ≥ 4**         | 63.0%                       |
> | **Keywords Scoring ≥ 5**         | 36.4%                       |
>
> The evaluation results demonstrate the quality of the Quest keywords' strong overall performance.
>
>
> **Q4: How is the oversampling ... Adding the mathematical formula for this would be helpful for the reader's understanding.**
>
> **A4:**  Essentially, oversampling redistributes the sampling probability to prioritize $I_s$, ensuring it gets a larger share of the total samples. We provide some explanations and mathematical formulas below:
>
> Assuming $ I_s$ contains $ n_s $  training samples, $ I_l$ contains $n_l$ training samples, $p$ is the oversampling probability, and the total number of training samples to be used is $N$. The number of samples drawn from $I_s$ can be calculated as:
>
> $ I_s =  \lceil \left( \frac{n_s}{n_s + n_l} + p \right) \cdot N \rceil $
>
> The number of samples drawn from $I_l$ can be calculated as:
>
> $ I_l = N - I_s$
>
> **Q5: What is the motivation for splitting into only two sets and instead of having a more fine-grained approach?**
>
> **A5:** As we mentioned in lines 222-224: ... the number of documents associated with different keywords varies significantly...we implement oversampling for documents with less frequent keywords." We also see the potential for Quest to combine with adaptive and more fine-grained approaches, and we will explore them in the future.
>
> **Q6: Figure 3 contains several typos: "query genaration" -> "query generation", "random smaple" -> "random sample"**
>
> **A6:** We apologize for the typos; we corrected them in the latest vision. Thank you for pointing it out.
>
> **Q7: In the results on Longbench in Table 1... More detailed explanations would be appreciated.**
>
> **A7:** Please refer to the General Author Response. We conducted a more comprehensive evaluation of QUEST using the latest long-context evaluation benchmark, HELMET. The performance across a broader range of subtasks statistically demonstrates that QUEST is effective.
>
> **Q8: It is surprising that Quest achieves 100% accuracy ... Are there any intuitive explanations for this phenomenon?**
>
> **A8:** We mentioned it in lines 371-376: Figure 1 is "retrieving a text sentence," and Figure 4 is "retrieving a numeric string." In addition, we use italics to highlight the difference in Figure 4's caption: "Unlike Figure 1 ... numeric string."

---

> > ### Comment · Reviewer_YyBr · 2024-11-22
> > **Response to authors**
> >
> > Thank you for taking the time to provide a detailed response.
> >
> > Q1: While I could understand the motivation of using real queries if these were available, generating a query for each article still seems fairly artificial and unjustified in comparison to using any summarization technique.
> >
> > Q2: Thank you for the clarification, it indeed seems reasonable that the random keyword assignment adopted in Quest is able to promote diversity in the context. It might have still been interesting to check the results when the keyword with the highest RAKE score is used, but I can understand that this would be a bit computationally intensive for the time frame of the discussion period.
> >
> > Q3: Thank you for this additional analysis on keyword quality, I think this is a nice addition to the paper. Please make sure to provide the details of this study (prompt for GPT-4o, information and instructions to the human participants, etc.) in the final version of the paper.
> >
> > Q4: Thank you for providing the details of the oversampling formula.
> >
> > Q5: Investigating more fine-grained strategies to split $I$ would be interesting, but I agree this may warrant another work.
> >
> > Q7: I appreciate the additional experiments on the HELMET dataset, although my question was more related to the positive impact of the Levenshtein distance metric on Quest.
> >
> > Q8: Thank you for pointing this out, I had missed this information.

---

> ### Author Response · Authors · 2024-11-25
> **Response to Reviewer YyBr**
>
> We sincerely appreciate your response! Your detailed and insightful feedback plays a crucial role in improving our article. The following text further clarifies some questions.
>
> **Q1: While I could understand the motivation... still seems fairly artificial and unjustified in comparison to using any summarization technique.**
>
> **A1:** We utilize the recently introduced Llama-3.2-1B model to generate summaries for 1,000 randomly sampled articles to analyze the differences between summarization and query generation. The results are as follows:
>
> - **Computational Cost**
>
> On average, each summary contains 147 characters, compared to 49 characters per query. The length gap indicates that generating summarizations requires roughly three times the computational resources needed to generate queries, negatively impacting this approach's scalability.
>
> - **Performance**
>
> Following A3, we use GPT-4o to evaluate the keywords extracted from queries and summarizations. The results are presented below:
> | **Metric**         | **Keywords from Queries** | **Keywords from Summaries** |
> |--------------------------------------|---------------------------|----------|
> | **Average Score**                    | 3.574                     | 3.009    |
> | **Keywords Scoring ≥ 3**         | 74.3%                     | 61.8%        |
> | **Keywords Scoring ≥ 4**         | 63.0%                       | 48.7%        |
> | **Keywords Scoring ≥ 5**         | 36.4%                       | 14.5%        |
>
> The results show that summaries' keywords are of less quality than those derived from queries.
>
> These results demonstrate that generating summaries incurs higher computational costs, resulting in lower-quality keyword extraction. We attribute this to the broader semantic space of summaries compared to queries, which interfere with effective keyword extraction.
>
> **Q2: ... It might have still been interesting to check the results when the keyword with the highest RAKE score is used ...**
>
> **A2:** Through entropy analysis, We evaluate the diversity of two strategies using the highest RAKE score and random sampling (scores greater than 3). The results below show that random sampling led to greater diversity.
>
> | **Metric**                           | **Highest RAKE Score** | **Random Sampling** |
> |--------------------------------------|---------------------------|----------|
> | **entropy score**                    | 15.31                     | 18.12    |
>
>
> We further construct a training dataset based on the highest RAKE score, and the results are presented in the table below:
> | Methods  | **AVG**   | **Recall** | **RAG**   | **ICL**   | **Re-rank** | **LongQA** |
> |----------|-------|--------|-------|-------|---------|--------|
> | Highest RAKE Score   | 49.51    | 66.76    | 56.27    | **72.72**    | 22.32    | 29.47    |
> | Random Sampling    | **50.97** | **69.13**  | **57.47** | 72.08 | **22.35**   | **33.82**  |
>
> These results demonstrate that random sampling enhances diversity and improves model performance.
>
>
> **Q7: ... although my question was more related to the positive impact of the Levenshtein distance metric on Quest.**
>
> **A7:** We apologize for misunderstanding your question.
>
> We observe that Levenshtein distance provides a more precise reflection of changes in long-context modeling capabilities compared to metrics like ROUGE. As a fine-grained metric, Levenshtein distance measures character-level differences more continuously, allowing for smoother tracking of improvements in handling long texts.
>
> Similar observations were mentioned in the recent HELMET long text evaluation paper and are supported by previous studies [1][2][3] that highlighted the noise issues of metrics such as ROUGE. HELMET [4] advocates for more precise metrics, such as exact matching and model-based evaluation, to assess long-context modeling performance.
>
> Combining our paper's experimental results with the HELMET results, it becomes clear that *Quest not only achieves strong performance on the Levenshtein distance metric but also shows improvement on other evaluation metrics*. This demonstrates that Quest effectively enhances long-context modeling capabilities.
>
> [1]Goyal T, Li J J, Durrett G. News summarization and evaluation in the era of gpt-3.
>
> [2]Deutsch D, Dror R, Roth D. Re-examining system-level correlations of automatic summarization evaluation metrics.
>
> [3]Chang Y, Lo K, Goyal T, et al. Booookscore: A systematic exploration of book-length summarization in the era of llms.
>
> [4]Yen H, Gao T, Hou M, et al. Helmet: How to evaluate long-context language models effectively and thoroughly.

---

> > ### Comment · Reviewer_YyBr · 2024-11-25
> > **Response to authors**
> >
> > Many thanks for your response and for the additional experiments. The comparison with the results for keywords extracted from summaries and for highest RAKE score keywords looks convincing, and it provides a nice empirical validation for QUEST's design choices. My main concerns have been addressed, so I will upgrade my rating to 8.

---

> ### Author Response · Authors · 2024-11-25
> **Supplementary Response  to Reviewer YyBr**
>
> **Q3: ...Please make sure to provide the details of this study (prompt for GPT-4o, information and instructions to the human participants, etc.) in the final version of the paper.**
>
> **A3:** We hired two PhD students to perform the annotation, along with GPT-4o, all following the scoring criteria:
>
> **Scoring Criteria**
> - **Score 1**: The keyword has little to no relevance to the article and may have been extracted incorrectly or is significantly off-topic.
> - **Score 2**: The keyword has some relevance to the article but is of low importance or overly generalized. It is not a core theme or focus of the article.
> - **Score 3**: The keyword has a clear connection to the article but is not one of its primary concepts or focal points.
> - **Score 4**: The keyword is strongly related to the article and represents an important concept or secondary theme.
> - **Score 5**: The keyword perfectly reflects the article's central theme, main concepts, or key discussion points.
>
> **Score distribution**
> | **Score** | **Keywords from Queries** | **Keywords from Summaries** |
> |-----------|---------------------------|-----------|
> | **5**     | 364                       |  145 |
> | **4**     | 266                       | 342 |
> | **3**     | 113                       | 131 |
> | **2**     | 94                        | 141 |
> | **1**     | 163                       | 241 |

---

### Official Review · Reviewer_W5Zm · 2024-11-02

**Soundness:** 2
**Presentation:** 2
**Contribution:** 3
**Rating:** 8
**Confidence:** 3

**Summary:**

This paper introduces Quest, a query-centric data synthesis approach addressing data scarcity and domain imbalance issues in long-context modeling of LLMs. Quest achieves a balance between semantic correlation and context diversity by predicting potential queries for each document and aggregating semantically relevant but low-redundancy documents based on query similarity and keywords. Experiments demonstrate Quest's superior performance on long-context tasks ranging from 32k to 1M tokens.

**Strengths:**

1. The motivation is clear. The authors identifie two critical challenges in long-context training data: scarcity and domain distribution imbalance, then provide analysis of limitations in existing methods (Standard, KNN, ICLM).
2. The authors propose a query-centric data synthesis approach to alleviate the above problem, includes: query-based document aggregation moduls, keyword extraction and filtering modules, split ratio mechanism.
3. Extensive evaluation across various model scales (1.4B-12B) and context lengths (32k-1M) supports the method's effectiveness.

**Weaknesses:**

1. The author used a query generation model to generate different queries for different documents, which would make the quality of query generation affect the final performance of the model. The author should discuss in more detail the impact of query quality or different query generation methods on model performance.
2. The model approach is more like an integration of existing tools, without any contribution to methodological or theoretical innovation. For example, the query generation and selection process directly employ the off-the-shelf tools (doc2query, RAKE) without significant enhancement.

**Questions:**

Please see the weaknesses.

---

> ### Author Response · Authors · 2024-11-21
> **Response to Reviewer W5Zm**
>
> **Q1: The author used a query generation model to generate different queries for different documents, which would make the quality of query generation affect the final performance of the model. The author should discuss in more detail the impact of query quality or different query generation methods on model performance.**
>
> **A1:** This is a very good suggestion. We attempted to generate queries using LLMs, which intuitively could produce better queries. However, the enormous computational cost exceeded what we could afford. To explore the quality of query generation within a manageable range, we randomly replaced keywords in the queries with a certain probability to simulate the creation of low-quality queries. The experimental results are as follows:
>
> | Replacement Ratio | AVG    | Recall  | RAG          | ICL    | Re-rank    | LongQA       |
> |-------------------|--------|---------|--------------|--------|------------|--------------|
> | 0%             | 50.97  | 69.125  | 57.47  | 72.08  | 22.35 | 33.82  |
> | 20%            | 50.33  | 67.35   | 58.07  | 71.72  | 22.55 | 31.98   |
> | 50%            | 49.74  | 63.96 | 58.19  | 73.32  | 20.93 | 32.31  |
> | 100%(Standard)   | 48.08 | 62.33  | 58.67 | 71.24 | 19.18  | 28.99  |
>
> The results show that the model's performance gradually declines as the replacement ratio increases, indicating that improving the quality of the query may further enhance performance.
>
>
> **Q2: The model approach is more like an integration of existing tools, without any contribution to methodological or theoretical innovation. For example, the query generation and selection process directly employ the off-the-shelf tools (doc2query, RAKE) without significant enhancement.**
>
> **A2:** Since the pre-training corpus is very large, to ensure the scalability of the Quest method, we mainly selected some lightweight components that had been verified to be reliable. The experimental results also confirm that our choice ensures both effectiveness and scalability. Although we have also explored some resource-intensive methods, such as using LLM to generate better queries, these methods are not practical at present because of their large resource consumption and inability to scale.

---

> > ### Comment · Reviewer_W5Zm · 2024-11-24
> >
> > Thanks for the authors' reponses. The author answered my doubts well. I will improve my rating: 5--->6.

---

> > > ### Comment · Reviewer_W5Zm · 2024-12-03
> > >
> > > I read the revised version, other reviewer's comments, and the author's responses. The author did very thorough experiments and ablations, and these conversations gave me greater interest and confidence in this paper. I decide to raise the rating to 8 points.

---

### Official Review · Reviewer_9HBG · 2024-11-05

**Soundness:** 3
**Presentation:** 3
**Contribution:** 2
**Rating:** 6
**Confidence:** 4

**Summary:**

This paper introduces QUEST, a long-context scaling method for LLM pretraining. The proposed method focuses on data sythesis and introduces a simple algorithm QUEST, in which queries are first generated by documents, followed by key words generation of the queries. Then, the keywords are used as inverted index to group similar documents, where the similar documents are arranged within the same training batch to improve long-context modeling by cross-document reasoning. The authors perform experiment on continual pretraining and demonstrate the effectiveness of the proposed QUEST on both long- and short-context benchmarks. The authors also provide additional analysis on the training dynamics and scaling properties of long-context modeling using QUEST.

**Strengths:**

1. The authors propose a simple and scalable method to group documents for long-context pretraining
2. The proposed method seem to work and achieve performance gains on the selected short- and long-context benchmarks
3. The authors also provide many interesting observations and analysis of QUEST and long-context modeling, which may be helpful for LLM pretraining

**Weaknesses:**

1. Although the proposed method is simple, it largely follows previous approaches but with a different indexing and sampling method. In addition, no ablation is provided so we are not sure which part of the QUEST is the most helpful (query-based indexing or sampling etc.).
2. The performance of QUEST is quite similar to ICLM on long-context benchmarks. Interestingly, in appendix, it seems QUEST often underperforms compared to ICLM or even KNN. THis raises questions on the real performance of the proposed method.
3. The authors provide many interesting observations and analysis, yet these do not necessarily demonstrate the validity of QUEST. For instance, the examples in 5.2 show that most baseline methods also have good clustering quality.

**Questions:**

1. In fig3, Llama 3 8B 128k has accuracy of 0.97, so why even longer context of 1M as in fig 1 would result in 1 accuracy?
2. Why the performance in appendix from table 8 to table 10 are in consistent, I find it hard to inteprete these results and how do they differ from the tasks that are present in the main text?

---

> ### Author Response · Authors · 2024-11-21
> **Response to Reviewer 9HBG**
>
> **Q1: Although the proposed method is simple, it largely follows previous approaches but with a different indexing and sampling method.**
>
> **A1:** Since the pre-training corpus is very large, to ensure the scalability of the Quest method, we mainly selected some lightweight components that had been verified to be reliable. The experimental results also confirm that our choice ensures both effectiveness and scalability. Although we have also explored some resource-intensive methods, such as using LLM to generate better queries, these methods are not practical at present because of their large resource consumption and inability to scale.
>
>
> **Q2: In addition, no ablation is provided so we are not sure which part of the QUEST is the most helpful (query-based indexing or sampling etc.).**
>
> **A2:**
>
> **a. query-based indexing**
>
> We employed GPT-4 to evaluate the quality of keywords with and without query-based indexing across 1000 samples, using a scoring system ranging from 1 to 5. The results are presented in the following tables:
>
> | **Metric**                           | **Query-Based Indexing** | **without Query-Based Indexing** |
> |--------------------------------------|---------------------------|----------|
> | **Average Score**                    | 3.574                     | 2.634    |
> | **Keywords Scoring ≥ 3**         | 74.3%                     | 49.3%        |
> | **Keywords Scoring ≥ 4**         | 63.0%                       | 31.2%        |
> | **Keywords Scoring ≥ 5**         | 36.4%                       | 6.0%        |
>
> The results indicate that query-based indexing significantly improves the quality of the extracted keywords. This is evident from the higher average score (3.574 vs. 2.634) and the increased proportion of keywords scoring 3 or above (74.3% vs. 49.3%). Furthermore, the percentage of keywords scoring 4 or 5 is notably higher with query-based indexing (63% vs. 31.2%).
>
> **b.sampling**
>
> In Appendix A.4, we examine the impact of the split ratio, which controls the proportion of oversampled keyword-based inverted indexes. As stated in line 790, "It suggests that moderate oversampling of indexes with fewer documents benefits long-context modelling, highlighting the importance of balanced data distribution for long-context tasks."
>
> **Q3: ... THis raises questions on the real performance of the proposed method.**
>
> **A3:** Please refer to the General Author Response. We conducted a more comprehensive evaluation of QUEST using the latest long-context evaluation benchmark, HELMET. The performance across a broader range of subtasks (as many as 17) statistically demonstrates that QUEST is sufficiently effective.
>
> **Q4: ...For instance, the examples in 5.2 show that most baseline methods also have good clustering quality.**
>
> **A4:** The validity of QUEST derives from the fact that Quest balances context diversity and semantic correlation. We demonstrate that **the methods overemphasizing semantic correlation (good clustering quality) hurt the performance**. As we mentioned in lines 88-92:  "The results show that either prioritizing context diversity at the expense of semantic correlation (Standard) or overemphasizing semantic correlation while sacrificing context diversity (KNN and ICLM) leads to suboptimal performance... highlighting the need for a method to balance both aspects."
>
> We also provide analysis in lines 474-478:  "Figure 5 (left) shows that the Standard method forms the most dispersed clusters due to random aggregation, while KNN and ICLM yield tighter clusters in the 32k-context-length setting ... Figure 5 (right) illustrates document distribution for various methods under the 128k context length. Quest continues to aggregate relevant, lowredundancy documents."
>
> **Q5: In fig3, Llama 3 8B 128k has accuracy of 0.97, so why even longer context of 1M as in fig 1 would result in 1 accuracy?**
>
> **A5:** Fig 1 and 3 are two different Needle-in-a-Haystack settings. We mentioned it in lines 371-176: Figure 1 is "retrieving a text sentence," and Figure 4 is "retrieving a numeric string."  In addition, we use italics to highlight the difference in Figure 4's caption: "Unlike Figure 1, task difficulty is increased within the 128k context length by retrieving a sentence rather than a random numeric string." They are both widely used, so we tested two datasets.
>
> **Q6: Why the performance in appendix from table 8 to table 10 are in consistent, I find it hard to inteprete these results and how do they differ from the tasks that are present in the main text?**
>
> **A6:** The Longbench benchmark aggregates many subtasks into different task categories. For example, the Single-Doc QA task category includes three subtasks: narrativeqa, qasper, and multifieldqa_en. Due to space limitations, we report the average performance of the aggregated task categories in the main experiment. In the appendix, we show the performance of each subtask.

---

> ### Author Response · Authors · 2024-11-24
> **Looking Forward to the Response from Reviewer 9HBG**
>
> Dear Reviewer 9HBG,
>
> We sincerely appreciate your thoughtful feedback, which has played a crucial role in improving the quality of our work. We have carefully prepared detailed responses for your review.
>
> Please don’t hesitate to let us know if further clarification is needed.
>
> Best regards,
>
> The Authors

---

> > ### Author Response · Authors · 2024-11-28
> > **Reminder for paper discussion**
> >
> > Dear Reviewer 9HBG,
> >
> > We uploaded modifications to the paper, and the updated version includes the following changes:
> >
> > In **Appendix C**, we provided the test results for HELMET and the ablation results for different steps of Quest. These results demonstrate that each step of Quest contributes positively to the model's performance.
> >
> > We would like to hear your thoughts on whether our response and the additional results address your concerns. Please feel free to share any further questions or suggestions you might have.
> >
> > Best regards,
> >
> > The Authors

---

> > > ### Comment · Reviewer_9HBG · 2024-12-03
> > > **Response to Rebuttal**
> > >
> > > Thank you for the detailed response and additional experimental results. Some of my concerns are addressed but the rest remains (e.g., limited improvements compared to existing methods). Considering these, I am raising the overall score to 6 with weak accept.

---

### Official Review · Reviewer_ki5C · 2024-11-05

**Soundness:** 2
**Presentation:** 3
**Contribution:** 2
**Rating:** 3
**Confidence:** 2

**Summary:**

This paper proposes a method to generate synthetic long documents for model training. It aims to solve the problem of standard method that may concatenate unrelated short document to form a long document, and the similarity-based methods that concatenate too similar documents leading to the lack of diversity. The proposed method - Quest - selects the documents to be concatenated by the common keywords they would have in the generated queries to which they may answer. This is intended to concatenate documents that may be on a common topic (keywords), without being too similar.
The experiments on several datasets show that the synthetic data generated by the proposed method can lead to some improvements in the test tasks (on average).

**Strengths:**

1. The problem observed in the previous methods to generate synthetic data is interesting: the documents to be concatenated should be similar to some extent, but not too similar.
2. The experiments show that the method can result in improvement on some tasks.

**Weaknesses:**

1. The paper proposes a quite complex algorithm using doc2query, keyword indexing, then sampling through keywords. The paper does not motivate why these steps are necessary. One could imagine that much simpler methods would be able to lead to synthetic data that are to some extent similar, as is required by the authors. For example, if it is observed that the method based on KNN lead to concatenating too similar documents, it would be possible to select documents to be concatenated that are similar to some extent, but not the most similar. Could this simpler method lead to similar results?
2. The paper tends to over-claim the advantage of the proposed method. For example, it is said "Table 1 compares the Longbench results, showing that Quest consistently outperforms other methods across model sizes, ...". Looking at Table 1, one can see that the proposed method outperforms the others on average, but underperform them on several datasets. The advantage of the method is not so clear. This over-claim appear in other observations as well (e.g. Table 3 shows that Pythia obtains the best performance on 4 datasets, while Quest only on 2). Globally, if one consider all the datasets, the demonstration that the proposed method is better is very weak.
3. Some of the figures are unclear. Fig. 1 is not well explained. It intends to show the perfect performance of Quest on a specific task. However, the figure does not provide very useful information in addition to say it is perfect. One would also like to see some other methods in comparison, and to understand better the task itself. In Fig. 2, the dotted lines are not explained. One can guess later that they correspond to the performance of the model in some task.
4. Fig. 5 compares the results using the documents of some length in the original dataset, and the synthetic documents. It shows that the latter perform slightly better. Do these data have equivalent sizes? If the synthetic data are much more than the real subset of data, the observation would not be surprising. However, it would be difficult to conclude that the synthetic data are better than the real data, only they are more.

**Questions:**

Have you tried simpler methods to select documents to be concatenated, e.g. based on controlled similarity measure, so that the documents are within some range of similarity? Would this achieve the same goal as the proposed method?

---

> ### Author Response · Authors · 2024-11-21
> **Response to Reviewer ki5C**
>
> **Q1: The paper does not motivate why these steps are necessary.**
>
> **A1:** As we mentioned in line 095: "Our inspiration stems from the observation that similar queries can aggregate semantic relevant but low-redundancy documents via search engines ...we predict potential queries using a generative model for each document in the training dataset." Our method, therefore, consists of three main steps: a. Query generation; b. keyword extraction; and c. index building:
>
> **a. Query generation**
>
> Our primary goal was to collect enormous queries and cluster articles related to each query, mimicking the functionality of search engines. However, collecting many queries is very time-consuming and cannot guarantee diversity. Thus, we employed the doc2query model to generate queries for each document in the corpus. This way, we solve the problem of cheaply acquiring many diverse queries.
>
> **b. Keyword extraction**
>
> However, we noticed that the generated queries were overly fine-grained, resulting in retrieved documents with insufficient diversity. To address this, we adopted a coarser-grained keyword-based approach for indexing, which not only improved diversity but also reduced redundancy.
>
> **c. Index building**
>
> By building an inverted index, we implemented an efficient aggregation pipeline, also the basis for scalable long-data synthesis.
>
> **Q2: One could imagine that much simpler methods would be able to lead to synthetic data that are to some extent similar ... Could this simpler method lead to similar results?**
>
> **A2:** This is a valuable hypothesis, and we designed three strategies to achieve the goal of "selecting documents to be concatenated that are similar to some extent."
>
> - **Strategy 1**: Documents with mid-ranking similarity were selected and concatenated.
> - **Strategy 2**: Documents were concatenated by random sampling.
> - **Strategy 3**: Documents were concatenated in reverse order of similarity.
>
> | Methods   | cosine similarity | AVG  | Recall | RAG  | ICL  | Re-rank | LongQA |
> |-----------|------------|------|--------|------|------|--------|--------|
> | KNN (topk)    | 60.97      | 46.31 | 64.24  | 56.00 | 60.28 | 18.77  | 32.27  |
> | KNN (mid-ranking) | 48.53      | 47.96 | 65.71| 56.97| 68.24 | 16.54 | 32.33|
> | KNN (random sampling) | 35.51      | 46.78 | 64.23  | 55.24 | 70.96 | 12.23  | 31.22  |
> | KNN (reverse order) | 35.49      | 46.18 | 62.19| 54.28| 66.28 | 17.20 | 30.93|
> | Quest     | 46.54      | **50.97** | **69.13**  | **57.47** | **72.08** | **22.35**  | **33.82**  |
>
> As shown in the table, for the KNN method, reducing the similarity of the selected documents results in a slight performance improvement, followed by a decline. Furthermore, these methods consistently perform worse than the Quest method.
>
> Additionally, while the "mid-ranking" approach exhibits a similarity extent close to the Quest method, its performance remains far inferior. It indicates that document aggregation has a greater impact on overall performance than the extent of similarity, highlighting it as the key factor driving Quest's superior results.
>
> **Q3: The paper tends to over-claim the advantage of the proposed method. Globally, if one consider all the datasets, the demonstration that the proposed method is better is very weak.**
>
> **A3:** Please refer to the General Author Response. We comprehensively evaluated QUEST using the latest long-context evaluation benchmark, HELMET. The results, covering up to 17 subtasks, statistically confirm QUEST's high effectiveness.
>
> **Q4: Table 3 shows that Pythia obtains the best performance on 4 datasets, while Quest only on 2.**
>
> **A4:** Table 3 demonstrates that training on long texts synthesized by Quest does not adversely affect the performance of short text tasks, aligning with the objectives of previous studies [1][2].
>
> While the performance of Quest on the four datasets is slightly lower than that of Pythia without long text training, the difference falls within the margin of evaluation fluctuations. Moreover, the average performance of Quest across multiple datasets is comparable to that of Pythia without long-text training, further supporting the conclusion that training on long texts synthesized by Quest does not compromise the performance of short-text tasks.
>
> [1] Dubey A, Jauhri A, Pandey A, et al. The llama 3 herd of models.
>
> [2] Fu Y, Panda R, Niu X, et al. Data engineering for scaling language models to 128k context.
>
> **Q5: Some of the figures are unclear.**
>
> **A5:** Thanks for pointing this out. We will improve and refine our figures in future versions.
>
> **Q6: Fig. 5 compares the results using the documents ... Do these data have equivalent sizes?**
>
> **A6:** Yes, we strictly controlled the amount of data in both the synthetic data and the long documents to be identical. We will provide more detailed explanations in the latest versions.
>
> I hope my responses answer your questions and increase your confidence in our paper. Thank you!

---

> ### Author Response · Authors · 2024-11-24
> **Looking Forward to the Response from Reviewer ki5C**
>
> Dear Reviewer ki5C,
>
> We sincerely appreciate the insightful feedback you provided, which has been instrumental in enhancing the quality of our work. We have prepared detailed responses for your review.
>
> Please let us know if we need to provide any further clarifications or feedback.
>
> Best regards,
>
> The Authors

---

> > ### Author Response · Authors · 2024-11-28
> > **Reminder for paper discussion**
> >
> > Dear Reviewer ki5C,
> >
> > We uploaded modifications to the paper, and the updated version includes the following changes:
> >
> > 1. In **Appendix C**, we provided the test results for HELMET and the ablation results for different steps of Quest. These results demonstrate that each step of Quest contributes positively to the model's performance.
> > 2. In **Appendix D.1**, we presented the results using different Knn strategies, highlighting the existing performance gap compared to Quest.
> > 3. We added a description of the dotted lines in Figure 2.
> > 4. We added the following statement to line 505:  *"The amount of data in both the synthetic data and the long documents is identical."*
> >
> > We would like to hear your thoughts on whether our response addresses your concerns. Please feel free to share any further questions or suggestions you might have.
> >
> > Best regards,
> > The Authors

---

### Author Response · Authors · 2024-11-21
**General Author Response**

We sincerely appreciate the thorough and thoughtful feedback from all reviewers! They confirmed the value of our interesting observations and analysis (ki5C, 9HBG), clear motivation (W5Zm), well-written presentation, and convincing experiments (YyBr).

---

However, some reviewers raised concerns about Quest's effectiveness. To address the concerns, we evaluate the Quest method on the newly proposed Helmet benchmark [1]. This benchmark provides more reliable and consistent rankings for frontier LCLMs. It spans evaluation lengths of 8k, 16k, 32k, 64k, and 128k across five task types and 17 subtasks, including Recall (4 subtasks), RAG (4 subtasks), ICL (5 subtasks), Re-rank (1 subtask), and LongQA (3 subtasks). The experimental results are summarized below:

| Methods  | **AVG**   | **Recall** | **RAG**   | **ICL**   | **Re-rank** | **LongQA** |
|----------|-------|--------|-------|-------|---------|--------|
| Standard | 48.08 | 62.33  | **58.67** | 71.24 | 19.18   | 28.99  |
| KNN      | 46.31 | 64.24  | 56.00 | 60.28 | 18.77   | 32.27  |
| ICLM     | 46.62 | 64.04  | 54.48 | 72.36 | 14.04   | 28.17  |
| Quest    | **50.97** | **69.13**  | 57.47 | **72.08** | **22.35**   | **33.82**  |

The results represent the average performance across lengths from 8k to 128k. The comprehensive evaluation on Helmet further shows that Quest achieves non-trivial improvements across multiple datasets, with a notable +2.89% average increase in performance on as many as 17 subtasks.

[1]Yen H, Gao T, Hou M, et al. Helmet: How to evaluate long-context language models effectively and thoroughly.

---

### Meta-Review · Area_Chair_NJcH · 2024-12-18

**Metareview:**

The paper introduces Quest, a query-centric data synthesis approach for improving long-context scaling of large language models (LLMs). The method emphasizes balancing semantic coherence and diversity within training data by grouping semantically relevant yet non-redundant documents. The authors empirically evaluate Quest on both synthetic and real-world long-context benchmarks. They also show that Quest does not compromise the performance of short-text tasks. The reviewers find that the approach is innovative (W5Zm, R8Kq), effective (R8Kq, Q2Lm), and scalable  (W5Zm, Q2Lm).

**Additional Comments On Reviewer Discussion:**

The authors have addressed reviewer concerns, including clarifying the keyword extraction process and providing additional results in the appendix (W5Zm, Q2Lm).

In the reviewer-AC discussion stage, reviewer ki5C remained concerned about the motivation and experimental demonstration of the paper. Other reviewers believe that these limitations are mostly addressed by the rebuttal, which the AC agrees with. However, the AC also agrees with reviewer ki5C that the claims on the experimental results should be adjusted, e.g., toning down a bit the statement on short text performances by mentioning there is "_almost_ no degradation in short text performance _on average_".  The authors are also encouraged to acknowledge that Quest is not optimal for some tasks and provide plausible explanations accordingly.

To sum up, Quest makes a positive contribution to the long-context LLM area. We recommend acceptance and urge the authors to make the requisite revisions pointed out by reviewers.

---

### Decision · Program_Chairs · 2025-01-22

Accept (Poster)